# The gut microbiota and depressive symptoms across ethnic groups

Jos A. Bosch [1,2] ✉, Max Nieuwdorp [3], Aeilko H. Zwinderman[4], Mélanie Deschasaux [4,5], Djawad Radjabzadeh[6], Robert Kraaij [6], Mark Davids [3], Susanne R. de Rooij[4,8] & Anja Lok [7,8]

The gut microbiome is thought to play a role in depressive disorders, which makes it an attractive target for interventions. Both the microbiome and depressive symptom levels vary substantially across ethnic groups. Thus, any intervention for depression targeting the microbiome requires understanding of microbiome-depression associations across ethnicities. Analysing data from the HELIUS cohort, we characterize the gut microbiota and its associations with depressive symptoms in 6 ethnic groups (Dutch, South-Asian Surinamese, African Surinamese, Ghanaian, Turkish, Moroccan; $N$ = 3211), living in the same urban area. Diversity of the gut microbiota, both within (α-diversity) and between individuals (β-diversity), predicts depressive symptom levels, taking into account demographic, behavioural, and medical differences. These associations do not differ between ethnic groups. Further, β-diversity explains 29%–18% of the ethnic differences in depressive symptoms. Bacterial genera associated with depressive symptoms belong to mulitple families, prominently including the families *Christensenellaceae, Lachnospiraceae*, and *Ruminococcaceae*. In summary, the results show that the gut microbiota are linked to depressive symptom levels and that this association generalizes across ethnic groups. Moreover, the results suggest that ethnic differences in the gut microbiota may partly explain parallel disparities in depression.

Depressive disorders affect an estimated 322 million people globally and are a leading cause of disability, mortality, and economic disparity[1,2]. Current treatment options are considered suboptimal and a more complete understanding of etiology, as well as the identification of effective interventional strategies, are urgently needed[3]. A promising novel development in this area pertains to the potential role of the gut microbiome, i.e., the diverse microbial communities living in the gut and their genetic material[4]. Research has demonstrated that the composition of the intestinal microbiota may impact cognition and affect through multiple pathways, collectively known as the gut-brain axis[4]. These novel insights have fueled the idea that modification of microbial ecology may provide new options for the treatment and prevention of depression[5].

At present, much of the supporting evidence still takes the form of extrapolations from non-human research, whereas the human data remains sparse and are mostly limited to small-scale studies[4] that yield

[1]Department of Psychology, University of Amsterdam, Amsterdam, the Netherlands. [2]Department of Medical Psychology, Amsterdam University Medical Centers, University of Amsterdam, Amsterdam, the Netherlands. [3]Department of Internal and Vascular Medicine, Amsterdam University Medical Centers, University of Amsterdam, Amsterdam, the Netherlands. [4]Department of Epidemiology and Data Science, Amsterdam University Medical Centers, University of Amsterdam, Amsterdam, the Netherlands. [5]Paris 13 – Sorbonne Paris Nord University, Inserm U1153, Inrae U1125, Cnam, Nutritional Epidemiology Research Team (EREN), Epidemiology and Statistics Research Center – University of Paris (CRESS), Bobigny, France. [6]Department of Internal Medicine, Erasmus Medical Center Rotterdam, Rotterdam, the Netherlands. [7]Department of Psychiatry, Amsterdam University Medical Centers, University of Amsterdam, Amsterdam, the Netherlands. [8]These authors contributed equally: Susanne R. de Rooij, Anja Lok. ✉e-mail: j.a.bosch@uva.nl

inconsistent findings[6–9]. While disappointing, such inconsistency might be expected in light of the complex composition of the gut microbiota, which is shaped by hundreds of bacterial species that exhibit a marked diversity across groups and individuals[10–16]. This complexity only adds to the similarly multi-determined and heterogeneous nature of depression[17]. Notably missing from the literature, therefore, are adequately powered studies in well-characterized populations that would allow more rigorous analyses of individual differences.

Two large-scale population studies have been published that would appear suitable to address the above issues[12,13]. The LifeLines Study (*N* = 1135) showed that depression (based on self-reported diagnosis) is significantly associated with β-diversity, indicating that depressed individuals have a microbiota composition that is distinguishable from those without depression[12]. The Flemish Gut Flora Project (*N* = 1068), in which a diagnosis of depression was obtained from physician records[13], replicates this association while adjusting for age, sex, BMI, and gastro-intestinal parameters. Further, after excluding participants on antidepressive medication and cross-validation in a separate cohort (i.e., the aforementioned LifeLines Study)[13], the study identified two genera (*Dialister* and *Coprococcus*), both belonging to the phylum Firmicutes, that were less abundant among those depressed. Whilst lending credibility to the idea of links between depression and the gut microbiome, both studies applied sparse confounder adjustments, e.g., related to lifestyle and health. Thus, uncertainties remain as to the exact interpretation of the microbiome-depression associations, which limits further progress towards diagnostic and clinical applications[4].

An additional issue is that prior epidemiological associations are established in ethnically homogenous populations of North-European ancestry[12,13]. Demographic factors probably represent the largest source of individual variation in the gut microbiome[11,12,14,15,18]. For example, analyses of a large epidemiological survey (Healthy Life in an Urban Setting study, HELIUS) showed that ethnicity explained far more of the differences in gut microbiota than any of the other measures collected, which included other demographic factors (e.g., age, sex), lifestyle factors, and medical information[14]. It is unknown to what extent microbiome-depression associations generalize across ethic groups and this, too, limits interpretation, especially when considering the parallel and substantial ethnic disparities in depression[19].

In light of the preceding discussion, the present study investigated associations between gut microbiota and depressive symptom levels in a large (*N* = 3021) multi-ethnic cohort (the HELIUS study), comprised of six ethnic groups living in the same urban geographic area[14,20,21]. The primary aim was to identify which taxonomic features of the gut microbiota are linked to depressive symptom levels, while adjusting for possible confounding by demographic, lifestyle, and medical factors. For most individuals depression is transient, with a median duration of three to six months. Auxiliary analyses therefore also took pre-existing markers of depression risk into account, as these may provide a window on the temporal specificity of associations between the microbiota and current symptom levels; these included prior depressive episodes, parental history of depression, and the personality trait neuroticism (a generic risk marker for psychopathology)[22]. The second aim was to determine if microbiota-depression associations generalize across ethnic groups. Such generalizability would greatly broaden the potential applicability of microbiome-based diagnostics and interventions[23]. Finally, the present study aimed to assess if ethnic differences in gut microbiota may account for ethnic disparities in depression. A parallel study[24] provides a large-scale epidemiological investigation of the relation between fecal microbiota and depressive symptoms among subjects of European ancestry, cross-validating data from the Amsterdam HELIUS cohort and the Rotterdam Study cohort.

## Results

After applying exclusion criteria (see description in "Methods" section) and accounting for occasional missing data, a total of between *N* = 3211 (Regression Model 1) and *N* = 3088 (Regression Model 3) participants were available for analyses. Table 1 provides summary data of the study sample and the main covariates.

### Alpha-diversity predicts depressive symptoms

As shown in Table 2, the Shannon Index predicted PHQ-9 depressive symptom scores in linear regression analyses. Inclusion of demographic covariates (Model 1: age, sex, ethnicity, education) substantially attenuated the association between the Shannon Index and the PHQ-9 sum scores (standardized $\beta = -0.0738$, $p < 0.001$) while improving the overall model fit ($\Delta R^2 = 0.0597$, $p < 0.001$; total $R^2 = 0.0736$). Ethnicity had by far the largest contribution to this model fit: after adjustment for sex and age the contribution of ethnicity was $\Delta R^2 = 0.0431$ ($p < 0.001$), with a modest additional impact of education ($\Delta R^2 = 0.0015$, $p = 0.024$). After sequentially adding lifestyle factors (Model 2: $\Delta R^2 = 0.0087$, $p < 0.001$) and medical variables (Model 3; $\Delta R^2 = 0.0267$), the Shannon index continued to predict depressive symptom scores (standardized $\beta = -0.0597$, $p = 0.001$ and $-0.0422$, $p = 0.023$, respectively). No significant ethnicity by alpha-diversity interaction was detected in any of the three models (Model 1; $p = 0.232$; Model 2; $p = 0.134$; Model 3; $p = 0.325$), indicating that the association between alpha diversity and depressive symptoms did not differ across ethnic groups. Also, when results were stratified per ethnic group, the $I^2$ consistently approximated zero (see Supplementary Fig. 1). Repeating the above analyses for the Simpson index yielded comparable results (see Supplementary Table 1).

To estimate the specificity of the above associations, analyses were repeated while adjusting for parental history of depression, number of prior depressive episodes, and neuroticism. α-diversity no longer significantly predicted depressive symptoms after adjustment for neuroticism. Conversely, α-diversity was significantly associated with neuroticism after adjustment for depressive symptoms in all 3 regression models, indicating that the neuroticism was the stronger predictor. Parental history and the number of prior depressive episodes only minimally attenuated the associations with depressive symptoms (Model 3, standardized $\beta > -0.0384$, $p < 0.033$).

Table 3 additionally presents the results of linear regression analyses using α-diversity (Shannon) as the outcome, i.e., reversing X and Y. The fully adjusted model (Model 3) explained approximately 18% of variance in α-diversity, which was mostly attributed to ethnicity ($\Delta R^2 = 0.1143$, $p < 0.001$, after inclusion of age and sex), whereby PHQ-9 scores remained a significant predictor of the Shannon index (see Table 3).

### Beta-diversity predicts depressive symptoms

The principal coordinates (Principal Coordinate Analyses, PCoA) derived from Bray-Curtis dissimilarity or weighted UniFrac distance matrices were entered as predictors in linear regression (with PHQ-9 sum scores as the dependent variable). Forward selection of the first 20 coordinates yielded 6 coordinates that compiled information predictive of depressive symptom scores, and these coordinates were used in subsequent regression analyses. Among these coordinates was PCoA #2, which predicted 6.50% (Bray-Curtis) and 9.73% (Weighted UniFrac) in microbiome composition. Notably, the multidimensional information compiled in this principal coordinate demonstrated a high correlation ($r = 0.83$) with the Shannon index, indicating that within this statistical approach (and contrary to how α-diversity is typically conceptualized) α-diversity is integral to β-diversity (see also Supplementary Fig. 2C).

Figure 1A shows that the 6 principal coordinates jointly explained between 1.5% ($\Delta R^2$ Model 1) and 0.5% ($\Delta R^2$ Model 3) of the

**Table 1 | Summary data of the study sample**

| | | | Dutch | South Asian Surinamese | African Surinamese | Ghanaian | Turkish | Moroccan | Total sample |
|---|---|---|---|---|---|---|---|---|---|
| | | N | 769 | 527 | 767 | 458 | 349 | 473 | 3211 |
| 1st generation? | Yes | N | | 489 | 703 | 450 | 288 | 408 | 2338 |
| | | % | | 92.8% | 91.7% | 98.3% | 82.5% | 86.3% | 90.8% |
| Gender | Female | N | 376 | 281 | 460 | 248 | 168 | 240 | 1773 |
| | | % | 48.9% | 53.3% | 60.0% | 54.1% | 48.1% | 50.7% | 53.0% |
| Age | Years | M | 52.1 | 52.9 | 52.8 | 48.5 | 45.1 | 46.8 | 50.4 |
| | | SD | 12.6 | 10.3 | 9.7 | 8.7 | 10.4 | 11.0 | 11.1 |
| Education | Primary or less | N | 26 | 110 | 41 | 145 | 116 | 167 | 605 |
| | | % | 3.4% | 21.0% | 5.4% | 32.0% | 33.6% | 35.6% | 18.2% |
| | Secondary | N | 135 | 204 | 319 | 177 | 82 | 99 | 1016 |
| | | % | 17.6% | 38.9% | 41.8% | 39.1% | 23.8% | 21.1% | 30.6% |
| | Vocational | N | 159 | 118 | 238 | 110 | 96 | 137 | 858 |
| | | % | 20.8% | 22.5% | 31.2% | 24.3% | 27.8% | 29.2% | 25.8% |
| | College or University | N | 445 | 92 | 166 | 21 | 51 | 66 | 841 |
| | | % | 58.2% | 17.6% | 21.7% | 4.6% | 14.8% | 14.1% | 25.3% |
| BMI | kg/m$^2$ | Mean | 25.2 | 26.9 | 28.9 | 28.3 | 28.7 | 28.4 | 27.4 |
| | | SD | 3.9 | 4.6 | 5.4 | 4.6 | 4.5 | 4.6 | 4.8 |
| Physical activity | Min/wk by | Mean | 2592 | 2636 | 2810 | 2619 | 2154 | 2360 | 2574 |
| | Intensity | SD | 1327 | 1791 | 1926 | 2225 | 1592 | 1715 | 1776 |
| | Meeting | N | 170 | 209 | 262 | 210 | 190 | 232 | 1273 |
| | WHO criteria?[a] | % | 22.1% | 39.7% | 34.2% | 45.9% | 54.4% | 49.3% | 38.1% |
| Smoker | Yes | N | 142 | 118 | 200 | 25 | 88 | 44 | 617 |
| | | % | 18.5% | 22.5% | 26.2% | 5.5% | 25.4% | 9.3% | 18.5% |
| Alcohol use | Yes | N | 701 | 273 | 501 | 212 | 80 | 27 | 1794 |
| | | % | 91.2% | 52.0% | 65.6% | 46.7% | 23.1% | 5.7% | 53.9% |
| GI disease | Yes | N | 31 | 36 | 36 | 8 | 24 | 39 | 174 |
| | | % | 4.0% | 6.8% | 4.7% | 1.7% | 6.9% | 8.2% | 5.2% |
| Diabetes | Yes | N | 52 | 172 | 133 | 80 | 48 | 74 | 559 |
| | | % | 6.8% | 32.6% | 17.4% | 17.5% | 13.8% | 15.6% | 16.7% |
| Antibiotics | Past 2 wks | N | 25 | 21 | 16 | 20 | 7 | 15 | 104 |
| | | % | 3.3% | 4.0% | 2.1% | 4.4% | 2.0% | 3.2% | 3.1% |
| PPI | Yes | N | 60 | 92 | 74 | 29 | 62 | 84 | 401 |
| | | % | 7.8% | 17.5% | 9.6% | 6.3% | 17.8% | 17.8% | 12.0% |
| Diarrhea | Past week | N | 76 | 64 | 89 | 15 | 34 | 55 | 333 |
| | | % | 10.3% | 12.7% | 12.0% | 3.4% | 10.3% | 11.9% | 10.4% |
| Alpha diversity | Shannon | Mean | 4.41 | 3.83 | 4.13 | 4.03 | 4.03 | 4.15 | 4.13 |
| | | SD | 0.40 | 0.47 | 0.44 | 0.47 | 0.49 | 0.52 | 0.50 |
| | Richness | Mean | 527.57 | 375.59 | 443.90 | 435.11 | 441.90 | 474.98 | 455.36 |
| | | SD | 106.74 | 114.31 | 108.36 | 102.17 | 102.65 | 115.85 | 118.95 |
| | Chao1 | Mean | 772.94 | 574.38 | 660.75 | 642.76 | 646.64 | 680.59 | 671.81 |
| | | SD | 772.94 | 574.38 | 660.75 | 642.76 | 646.64 | 680.59 | 175.39 |
| | PD | Mean | 36.20 | 26.54 | 31.18 | 30.27 | 31.28 | 33.90 | 31.87 |
| | | SD | 6.74 | 6.97 | 7.04 | 6.29 | 6.48 | 7.38 | 7.54 |

[a]At least 30 min of moderate exercise, at least 5 days a week.

variance in depression scores. The results presented in Fig. 1B further revealed that fecal microbial composition explained between 28% (Model 1) and 18% (Model 3) of the ethnic differences in depression symptom scores. The β-diversity coordinates still significantly predicted depressive symptoms after adjustment for parental history of depression, prior depressive episodes, or neuroticism (all analyses ($\Delta R^2 > 0.0036$. $p < 0.002$). Replicating these analyses using weighted UniFrac distances (instead of Bray-Curtis dissimilarity) yielded equivalent results. The source data of Fig. 1 can be found in Supplementary Data file 1.

**Most taxa associated with depressive symptoms are Firmicutes**
Shown in Fig. 2, out of 416 non-trivial ASVs, 117 showed a significant unadjusted correlations with PHQ-9 scores (Rho, FDR < 0.05), with most (99 ASVs) showing a negative correlation (indicating a relative depletion). The source data for Table 2 is provided in the supplement (Supplementary Data file 3), which shows a subset of data from Supplementary Data file 3 (Supplementary Data file 3 is presented in table format in Supplementary Data files 4, 5). Approximately 65% identified the phylum *Firmicutes* (76 ASVs). To circumvent excessive multiple testing, only significant associations obtained in unadjusted

**Table 2 | Results of linear regression models with depressive symptom scores as the dependent variable**

**A. Model summary**

| | R² | ΔR² | F change | df1 | df2 | p |
|---|---|---|---|---|---|---|
| Unadjusted | 0.0144 | | 44.31 | 1 | 3033 | <0.0001 |
| Model 1 | 0.0754 | 0.0610 | 24.95 | 8 | 3025 | <0.0001 |
| Model 2 | 0.0841 | 0.0087 | 7.19 | 4 | 3021 | <0.0001 |
| Model 3 | 0.1109 | 0.0267 | 18.13 | 5 | 3016 | <0.0001 |

**B. Overview of coefficients**

| Dependent: PHQ-9 sum score | Unadjusted | | | Model 1 | | | Model 2 | | | Model 3 | | |
|---|---|---|---|---|---|---|---|---|---|---|---|---|
| | B | t | p | B | t | p | B | t | p | B | t | p |
| Shannon | −0.1200 | −6.66 | 0.0000 | −0.0729 | −3.84 | 0.0001 | −0.0602 | −3.15 | 0.0016 | −0.0422 | −2.22 | 0.0263 |
| Gender (Women = 1) | | | | 0.1154 | 6.52 | 0.0000 | 0.1266 | 6.82 | 0.0000 | 0.1186 | 6.45 | 0.0000 |
| Age | | | | −0.0388 | −2.06 | 0.0397 | −0.0406 | −2.13 | 0.0333 | −0.0656 | −3.38 | 0.0007 |
| Education | | | | −0.0426 | −2.10 | 0.0360 | −0.0346 | −1.68 | 0.0922 | −0.0227 | −1.12 | 0.2627 |
| Smoker (Yes = 1) | | | | | | | 0.0380 | 2.04 | 0.0410 | 0.0395 | 2.15 | 0.0314 |
| Physical activity (min/wk*intensity) | | | | | | | −0.0440 | −2.49 | 0.0127 | −0.0371 | −2.13 | 0.0333 |
| Audit alcohol (sum) | | | | | | | 0.0614 | 2.99 | 0.0028 | 0.0588 | 2.90 | 0.0037 |
| BMI (kg/m²) | | | | | | | 0.0519 | 2.74 | 0.0062 | 0.0444 | 2.35 | 0.0188 |
| GI disorder (yes = 1) | | | | | | | | | | 0.1027 | 5.85 | 0.0000 |
| Diabetes (yes = 1) | | | | | | | | | | 0.0624 | 3.36 | 0.0008 |
| Diarrhea (yes = 1) | | | | | | | | | | 0.0799 | 4.57 | 0.0000 |
| PPI (yes = 1) | | | | | | | | | | 0.0394 | 2.19 | 0.0284 |
| Antibiotics (yes = 1) | | | | | | | | | | 0.0264 | 1.53 | 0.1267 |

Tables present a model summary (A) and an overview of standardized regression coefficients (β) of predictor and covariates. Unstandardized values and constants are omitted for clarity. Models 1 to 3 also include ethnicity. Coefficients for ethnicity are reported separately in the text.

**Table 3 | Results of linear regression models with Shannon index as the dependent variable**

**A. Model summary**

| | $R^2$ | $\Delta R^2$ | F change | df1 | df2 | p |
|---|---|---|---|---|---|---|
| Unadjusted | 0.0144 | | 44.31 | 1 | 3033 | <0.0001 |
| Model 1 | 0.1568 | 0.1423 | 63.86 | 8 | 3025 | <0.0001 |
| Model 2 | 0.1720 | 0.0155 | 13.91 | 4 | 3021 | <0.0001 |
| Model 3 | 0.1831 | 0.0112 | 8.15 | 5 | 3016 | <0.0001 |

**B. Overview of coefficients**

| Dependent: | | Unadjusted | | | Model 1 | | | Model 2 | | | Model 3 | | |
|---|---|---|---|---|---|---|---|---|---|---|---|---|---|
| Shannon index | | B | t | p | B | t | p | B | t | p | B | t | p |
| PHQ-9 | (sum-score) | -0.1200 | -6.66 | 0.0000 | -0.0664 | -3.84 | 0.0001 | -0.0544 | -3.15 | 0.0016 | -0.0388 | -2.22 | 0.0263 |
| Gender | (Women = 1) | | | | 0.0839 | 4.95 | 0.0000 | 0.0721 | 4.06 | 0.0000 | 0.0740 | 4.18 | 0.0000 |
| Age | | | | | 0.0571 | 3.17 | 0.0015 | 0.0663 | 3.66 | 0.0003 | 0.0768 | 4.13 | 0.0000 |
| Education | | | | | 0.1089 | 5.64 | 0.0000 | 0.0913 | 4.70 | 0.0000 | 0.0850 | 4.39 | 0.0000 |
| Smoker | (Yes = 1) | | | | | | | -0.0621 | -3.52 | 0.0004 | -0.0619 | -3.52 | 0.0004 |
| Physical activity | (min/wk*intensity) | | | | | | | 0.0078 | 0.46 | 0.6429 | 0.0064 | 0.38 | 0.7018 |
| Audit alcohol | (sum) | | | | | | | -0.0555 | -2.85 | 0.0044 | -0.0545 | -2.81 | 0.0050 |
| BMI | (kg/m²) | | | | | | | -0.1036 | -5.77 | 0.0000 | -0.0995 | -5.52 | 0.0000 |
| GI disorder | (yes = 1) | | | | | | | | | | -0.0318 | -1.88 | 0.0600 |
| Diabetes | (yes = 1) | | | | | | | | | | -0.0408 | -2.29 | 0.0219 |
| Diarrhea | (yes = 1) | | | | | | | | | | -0.0523 | -3.12 | 0.0018 |
| PPI | (yes = 1) | | | | | | | | | | 0.0001 | 0.00 | 0.9976 |
| Antibiotics | (yes = 1) | | | | | | | | | | -0.0709 | -4.29 | 0.0000 |

Tables present model summary (A) and overview of standardized regression coefficients (β) of predictor and covariates. Unstandardized values and constants are omitted for clarity. Models 1 to 3 also include ethnicity. Coefficients for ethnicity are reported separately in the text.

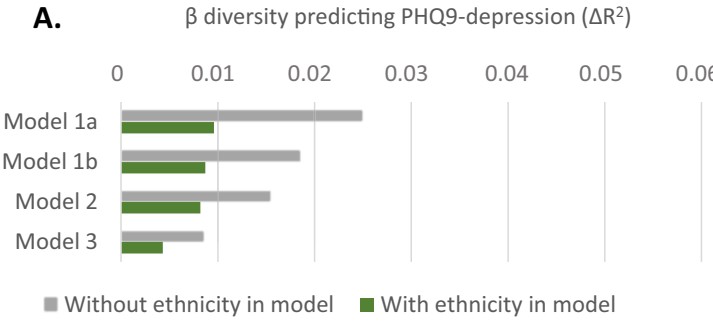

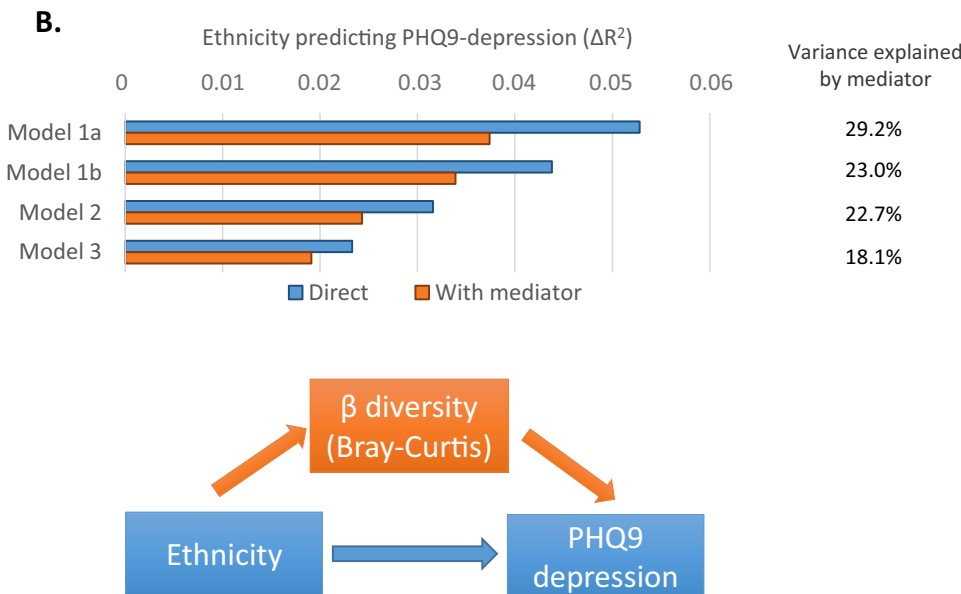

**Fig. 1 | Beta-diversity is linked with ethnic differences in depressive symptom scores. A** Beta-diversity predicting PHQ9 depression. It presents results of linear regression analyses that model β-diversity as a predictor of depressive symptom levels. Panel (**A**) horizontal bars present $\Delta R^2$ after progressive adjustments for confounders (models 1a to 3), and respectively without and with ethnicity included in each regression model. **B** Ethnicity predicting PHQ9 depression. It presents results of linear regression in which β-diversity is modeled as a mediator of the association between ethnicity and depressive symptom levels (see lower figure). Bars present $\Delta R^2$ the prediction of PHQ9 by ethnicity after progressive adjustments (models 1a to 3). Blue bars present $\Delta R^2$ without β diversity incorporated as a mediator in the model, and the orange bars present $\Delta R^2$ when mediation is assumed. The % in the table (right) indicate the attenuation of the direct effect by mediation. Regression models: We used two-sided linear regression analyses, no adjustments were made for multiple comparisons. Model 1a adjusted for age and gender; Model 1b added education; Model 2 further added behavioral factors (alcohol, smoking, exercise, BMI); Model 3 added GI disease, Diabetes, PPI use, Recent antibiotics, Diarrhea. All $\Delta R^2$ $p \leq 0.001$, except for ethnicity-inclusive Model 3 ($p = 0.023$).

analyses (FDR corrected) were further analyzed in subsequent Models 1–3 (using rank-transformed dependent Y). Figure 2 shows that 70 taxa remained significantly associated with PHQ-9 scores after adjustment for age, gender and ethnicity. The vast majority (60 ASVs) of these belonged to the phylum *Firmicutes*, with a prominent presence of the genus *Christensenellaceae* (group R7) and various genera within the families *Lachnospiraceae* (e.g., *Blautia, Lachnospiraceae* NK4A136, *Marvinbryantia, Roseburia*) and *Ruminococcaceae* (e.g., *Oscillibacter, Ruminicoccus* 1, Ruminococcaceae NK4A214 group, *Ruminococcaceae* UCG-005). Less prominent phyla included *Bacteroidetes* (e.g., genus *Bacteroides*) and *Proteobacteria* (genus *Desulfovibrio* and *Escherichia/Shigella*). Further adjustment for behavioral and medical variables (models 2 and 3) reduced the number of significant associations, yielding respectively 48 and 23 taxa that remained significantly associated with depression scores (see Fig. 2 and V).

The Supplementary Fig. 2 (panel A and B) provide a focused overview of correlations between PHQ-9 scores and individual ASVs (simultaneously plotted against correlations with alpha-diversity on the y-axis). Notable from these supplementary Figures, as well as from Fig. 2 (also see the corresponding source data), is that occasionally ASVs within the same genus showed opposite associations with depressive symptom scores, e.g., *Blautia, Bacteroides,* and *Oscillospira* (note that ASVs that the Greengenes database allocates to the single genus Oscillospira are attributed to multiple genera in the Sylva database, see the discussion). Whereas for other genera a more consistent pattern of associations was observed (e.g., *Christensenellaceae, Desulvofibrio, Streptococcus*).

Supplementary Data 4, 5 (and the corresponding Supplementary Data source data file 3) provide a heatmap depicting the correlations between individual ASVs, depressed mood and relevant depression risk factors and covariates. Heatmap inspection revealed that taxa that showed a strong correlation with depressive symptoms also tended to exhibit stronger correlations with selected covariates as well as with markers of alpha-diversity. As a further visualization, Supplementary Fig. 3 shows several examples of such associations in pair-wise scatterplots (these are based on the same source data files as Supplementary Data files 3, 4, 5).

| ASV_ID | Phylum | Class | Order | Family | genus | Species | Core | Beta Model 0 | p<.05 | Beta Model 1 | p<.05 | Beta Model 1 | p<.05 | Beta Model 3 | p<.05 |
|---|---|---|---|---|---|---|---|---|---|---|---|---|---|---|---|
| Zotu161 | Actinobacteria | Actinobacteria | Bifidobacteriales | Bifidobacteriaceae | Bifidobacterium | bifidum | | | ✓ | | | | | | |
| Zotu21 | Actinobacteria | Actinobacteria | Bifidobacteriales | Bifidobacteriaceae | Bifidobacterium | | ✓ | | ✓ | | | | | | |
| Zotu121 | Actinobacteria | Actinobacteria | Bifidobacteriales | Bifidobacteriaceae | Bifidobacterium | | | | ✓ | | | | | | ✓ |
| Zotu383 | Actinobacteria | Coriobacteriia | Coriobacteriales | Atopobiaceae | Libanicoccus | | | | ✓ | | | | | | |
| Zotu165 | Actinobacteria | Coriobacteriia | Coriobacteriales | Eggerthellaceae | Enterorhabdus | | | | ✓ | | ✓ | | | | |
| Zotu179 | Actinobacteria | Coriobacteriia | Coriobacteriales | Eggerthellaceae | | | | | ✓ | | ✓ | | ✓ | | |
| Zotu196 | Actinobacteria | Coriobacteriia | Coriobacteriales | Eggerthellaceae | Senegalimassilia | anaerobia | | | ✓ | | | | | | |
| Zotu218 | Bacteroidetes | Bacteroidia | Bacteroidales | Bacteroidaceae | Bacteroides | coprophilus | | | ✓ | | ✓ | | ✓ | | ✓ |
| Zotu70 | Bacteroidetes | Bacteroidia | Bacteroidales | Bacteroidaceae | Bacteroides | massiliensis | | | ✓ | | | | | | |
| Zotu55 | Bacteroidetes | Bacteroidia | Bacteroidales | Bacteroidaceae | Bacteroides | | | | ✓ | | | | ✓ | | ✓ |
| Zotu93 | Bacteroidetes | Bacteroidia | Bacteroidales | Bacteroidaceae | Bacteroides | | ✓ | | ✓ | | | | | | |
| Zotu205 | Bacteroidetes | Bacteroidia | Bacteroidales | Marinifilaceae | Odoribacter | splanchnicus | ✓ | | ✓ | | | | | | |
| Zotu272 | Bacteroidetes | Bacteroidia | Bacteroidales | Prevotellaceae | | | | | ✓ | | ✓ | | | | ✓ |
| Zotu308 | Bacteroidetes | Bacteroidia | Bacteroidales | Prevotellaceae | Paraprevotella | | | | ✓ | | | | | | |
| Zotu246 | Bacteroidetes | Bacteroidia | Bacteroidales | Rikenellaceae | Alistipes | | | | ✓ | | | | | | |
| Zotu66 | Bacteroidetes | Bacteroidia | Bacteroidales | Rikenellaceae | Alistipes | putredinis | | | ✓ | | | | | | |
| Zotu173 | Bacteroidetes | Bacteroidia | Bacteroidales | Rikenellaceae | Alistipes | shahii | ✓ | | ✓ | | | | | | |
| Zotu73 | Bacteroidetes | Bacteroidia | Bacteroidales | Tannerellaceae | Parabacteroides | merdae | ✓ | | ✓ | | | | | | |
| Zotu310 | Cyanobacteria | Melainabacteria | Gastranaerophilales | | | | | | ✓ | | | | | | |
| Zotu271 | Cyanobacteria | Melainabacteria | Gastranaerophilales | | | | | | ✓ | | ✓ | | ✓ | | |
| Zotu345 | Firmicutes | Bacilli | Lactobacillales | Lactobacillaceae | Lactobacillus | | | | ✓ | | ✓ | | ✓ | | ✓ |
| Zotu336 | Firmicutes | Bacilli | Lactobacillales | Streptococcaceae | Streptococcus | | | | ✓ | | ✓ | | ✓ | | |
| Zotu25 | Firmicutes | Bacilli | Lactobacillales | Streptococcaceae | Streptococcus | | ✓ | | ✓ | | ✓ | | ✓ | | |
| Zotu138 | Firmicutes | Clostridia | Clostridiales | Christensenellaceae | Christensenellaceae_R-7_group | | ✓ | | ✓ | | ✓ | | ✓ | | ✓ |
| Zotu42 | Firmicutes | Clostridia | Clostridiales | Christensenellaceae | Christensenellaceae_R-7_group | | ✓ | | ✓ | | ✓ | | ✓ | | |
| Zotu347 | Firmicutes | Clostridia | Clostridiales | Christensenellaceae | Christensenellaceae_R-7_group | | | | ✓ | | ✓ | | ✓ | | |
| Zotu166 | Firmicutes | Clostridia | Clostridiales | Christensenellaceae | Christensenellaceae_R-7_group | | | | ✓ | | ✓ | | ✓ | | |
| Zotu326 | Firmicutes | Clostridia | Clostridiales | Christensenellaceae | Christensenellaceae_R-7_group | | | | ✓ | | | | | | |
| Zotu405 | Firmicutes | Clostridia | Clostridiales | Christensenellaceae | Christensenellaceae_R-7_group | | | | ✓ | | ✓ | | | | |
| Zotu227 | Firmicutes | Clostridia | Clostridiales | Clostridiaceae_1 | Clostridium_sensu_stricto_1 | | | | ✓ | | | | | | |
| Zotu354 | Firmicutes | Clostridia | Clostridiales | Family_XIII | Family_XIII_AD3011_group | | ✓ | | ✓ | | | | | | |
| Zotu129 | Firmicutes | Clostridia | Clostridiales | Lachnospiraceae | Agathobacter | | ✓ | | ✓ | | | | | | |
| Zotu332 | Firmicutes | Clostridia | Clostridiales | Lachnospiraceae | Blautia | caecimuris | | | ✓ | | | | ✓ | | ✓ |
| Zotu207 | Firmicutes | Clostridia | Clostridiales | Lachnospiraceae | Blautia | | | | ✓ | | | | ✓ | | ✓ |
| Zotu416 | Firmicutes | Clostridia | Clostridiales | Lachnospiraceae | Blautia | | | | ✓ | | | | ✓ | | ✓ |
| Zotu34 | Firmicutes | Clostridia | Clostridiales | Lachnospiraceae | Blautia | obeum | ✓ | | ✓ | | | | ✓ | | ✓ |
| Zotu68 | Firmicutes | Clostridia | Clostridiales | Lachnospiraceae | Butyrivibrio | crossotus | | | ✓ | | | | | | |
| Zotu311 | Firmicutes | Clostridia | Clostridiales | Lachnospiraceae | GCA-900066575 | | ✓ | | ✓ | | ✓ | | ✓ | | ✓ |
| Zotu428 | Firmicutes | Clostridia | Clostridiales | Lachnospiraceae | Herbinix | | | | ✓ | | | | | | |
| Zotu276 | Firmicutes | Clostridia | Clostridiales | Lachnospiraceae | Lachnoclostridium | | | | ✓ | | ✓ | | ✓ | | ✓ |
| Zotu75 | Firmicutes | Clostridia | Clostridiales | Lachnospiraceae | Lachnospira | | ✓ | | ✓ | | | | | | |
| Zotu64 | Firmicutes | Clostridia | Clostridiales | Lachnospiraceae | Lachnospira | | ✓ | | ✓ | | | | | | |
| Zotu240 | Firmicutes | Clostridia | Clostridiales | Lachnospiraceae | Lachnospiraceae_AC2044_group | | | | ✓ | | | | | | |
| Zotu324 | Firmicutes | Clostridia | Clostridiales | Lachnospiraceae | Lachnospiraceae_FCS020_group | | | | ✓ | | ✓ | | ✓ | | ✓ |
| Zotu36 | Firmicutes | Clostridia | Clostridiales | Lachnospiraceae | Lachnospiraceae_ND3007_group | | ✓ | | ✓ | | ✓ | | ✓ | | |
| Zotu294 | Firmicutes | Clostridia | Clostridiales | Lachnospiraceae | Lachnospiraceae_NK4A136_group | | | | ✓ | | ✓ | | ✓ | | |
| Zotu86 | Firmicutes | Clostridia | Clostridiales | Lachnospiraceae | Lachnospiraceae_NK4A136_group | | ✓ | | ✓ | | ✓ | | | | |
| Zotu58 | Firmicutes | Clostridia | Clostridiales | Lachnospiraceae | Lachnospiraceae_NK4A136_group | | | | ✓ | | | | | | |
| Zotu401 | Firmicutes | Clostridia | Clostridiales | Lachnospiraceae | Lachnospiraceae_NK4A136_group | | | | ✓ | | | | | | |
| Zotu188 | Firmicutes | Clostridia | Clostridiales | Lachnospiraceae | Lachnospiraceae_UCG-001 | | | | ✓ | | ✓ | | ✓ | | |
| Zotu361 | Firmicutes | Clostridia | Clostridiales | Lachnospiraceae | Lachnospiraceae_UCG-003 | | | | ✓ | | ✓ | | | | |
| Zotu318 | Firmicutes | Clostridia | Clostridiales | Lachnospiraceae | Lachnospiraceae_UCG-008 | | | | ✓ | | | | | | |
| Zotu374 | Firmicutes | Clostridia | Clostridiales | Lachnospiraceae | Marvinbryantia | | | | ✓ | | | | ✓ | | |
| Zotu341 | Firmicutes | Clostridia | Clostridiales | Lachnospiraceae | Marvinbryantia | | | | ✓ | | ✓ | | ✓ | | ✓ |
| Zotu267 | Firmicutes | Clostridia | Clostridiales | Lachnospiraceae | Marvinbryantia | | ✓ | | ✓ | | ✓ | | ✓ | | ✓ |
| Zotu41 | Firmicutes | Clostridia | Clostridiales | Lachnospiraceae | | | ✓ | | ✓ | | ✓ | | ✓ | | ✓ |
| Zotu91 | Firmicutes | Clostridia | Clostridiales | Lachnospiraceae | | | ✓ | | ✓ | | ✓ | | ✓ | | ✓ |
| Zotu238 | Firmicutes | Clostridia | Clostridiales | Lachnospiraceae | | | ✓ | | ✓ | | | | | | |
| Zotu134 | Firmicutes | Clostridia | Clostridiales | Lachnospiraceae | | | | | ✓ | | | | | | |
| Zotu252 | Firmicutes | Clostridia | Clostridiales | Lachnospiraceae | | | ✓ | | ✓ | | ✓ | | | | |
| Zotu384 | Firmicutes | Clostridia | Clostridiales | Lachnospiraceae | | | | | ✓ | | | | | | |
| Zotu397 | Firmicutes | Clostridia | Clostridiales | Lachnospiraceae | | | ✓ | | ✓ | | | | | | |
| Zotu389 | Firmicutes | Clostridia | Clostridiales | Lachnospiraceae | | | | | ✓ | | ✓ | | | | |
| Zotu106 | Firmicutes | Clostridia | Clostridiales | Lachnospiraceae | | | | | ✓ | | ✓ | | | | |
| Zotu39 | Firmicutes | Clostridia | Clostridiales | Lachnospiraceae | Roseburia | inulinivorans | ✓ | | ✓ | | ✓ | | ✓ | | ✓ |
| Zotu141 | Firmicutes | Clostridia | Clostridiales | | | | | | ✓ | | ✓ | | | | |
| Zotu334 | Firmicutes | Clostridia | Clostridiales | Peptococcaceae | Peptococcus | | | | ✓ | | ✓ | | ✓ | | ✓ |
| Zotu133 | Firmicutes | Clostridia | Clostridiales | Ruminococcaceae | CAG-352 | | | | ✓ | | | | | | |
| Zotu5420 | Firmicutes | Clostridia | Clostridiales | Ruminococcaceae | Faecalibacterium | | ✓ | | ✓ | | | | | | |
| Zotu4 | Firmicutes | Clostridia | Clostridiales | Ruminococcaceae | Faecalibacterium | prausnitzii | ✓ | | ✓ | | | | | | |
| Zotu251 | Firmicutes | Clostridia | Clostridiales | Ruminococcaceae | Flavonifractor | plautii | | | ✓ | | ✓ | | | | |
| Zotu200 | Firmicutes | Clostridia | Clostridiales | Ruminococcaceae | Intestinimonas | | | | ✓ | | ✓ | | ✓ | | |
| Zotu245 | Firmicutes | Clostridia | Clostridiales | Ruminococcaceae | | | | | ✓ | | ✓ | | ✓ | | |
| Zotu322 | Firmicutes | Clostridia | Clostridiales | Ruminococcaceae | | | | | ✓ | | ✓ | | | | |
| Zotu168 | Firmicutes | Clostridia | Clostridiales | Ruminococcaceae | | | | | ✓ | | | | | | |
| Zotu208 | Firmicutes | Clostridia | Clostridiales | Ruminococcaceae | Oscillibacter | | ✓ | | ✓ | | ✓ | | ✓ | | ✓ |
| Zotu146 | Firmicutes | Clostridia | Clostridiales | Ruminococcaceae | Ruminiclostridium_5 | | ✓ | | ✓ | | | | | | |
| Zotu88 | Firmicutes | Clostridia | Clostridiales | Ruminococcaceae | Ruminiclostridium_6 | | | | ✓ | | | | | | |
| Zotu281 | Firmicutes | Clostridia | Clostridiales | Ruminococcaceae | Ruminiclostridium_9 | | | | ✓ | | ✓ | | ✓ | | |
| Zotu248 | Firmicutes | Clostridia | Clostridiales | Ruminococcaceae | Ruminiclostridium_9 | | ✓ | | ✓ | | ✓ | | | | |
| Zotu190 | Firmicutes | Clostridia | Clostridiales | Ruminococcaceae | Ruminococcaceae_NK4A214_group | | | | ✓ | | ✓ | | | | |
| Zotu385 | Firmicutes | Clostridia | Clostridiales | Ruminococcaceae | Ruminococcaceae_NK4A214_group | | | | ✓ | | ✓ | | | | |
| Zotu169 | Firmicutes | Clostridia | Clostridiales | Ruminococcaceae | Ruminococcaceae_NK4A214_group | | | | ✓ | | | | | | |
| Zotu35 | Firmicutes | Clostridia | Clostridiales | Ruminococcaceae | Ruminococcaceae_UCG-002 | bacterium | ✓ | | ✓ | | | | | | |
| Zotu37 | Firmicutes | Clostridia | Clostridiales | Ruminococcaceae | Ruminococcaceae_UCG-002 | | ✓ | | ✓ | | | | | | |
| Zotu114 | Firmicutes | Clostridia | Clostridiales | Ruminococcaceae | Ruminococcaceae_UCG-002 | | | | ✓ | | | | | | |
| Zotu184 | Firmicutes | Clostridia | Clostridiales | Ruminococcaceae | Ruminococcaceae_UCG-002 | | | | ✓ | | | | | | |
| Zotu301 | Firmicutes | Clostridia | Clostridiales | Ruminococcaceae | Ruminococcaceae_UCG-003 | | | | ✓ | | ✓ | | | | |
| Zotu143 | Firmicutes | Clostridia | Clostridiales | Ruminococcaceae | Ruminococcaceae_UCG-003 | | ✓ | | ✓ | | | | | | |
| Zotu110 | Firmicutes | Clostridia | Clostridiales | Ruminococcaceae | Ruminococcaceae_UCG-005 | | | | ✓ | | ✓ | | ✓ | | ✓ |
| Zotu392 | Firmicutes | Clostridia | Clostridiales | Ruminococcaceae | Ruminococcaceae_UCG-005 | | | | ✓ | | ✓ | | | | |
| Zotu162 | Firmicutes | Clostridia | Clostridiales | Ruminococcaceae | Ruminococcaceae_UCG-005 | | | | ✓ | | ✓ | | | | |
| Zotu351 | Firmicutes | Clostridia | Clostridiales | Ruminococcaceae | Ruminococcaceae_UCG-005 | | | | ✓ | | ✓ | | | | |
| Zotu67 | Firmicutes | Clostridia | Clostridiales | Ruminococcaceae | Ruminococcaceae_UCG-005 | | ✓ | | ✓ | | | | | | |
| Zotu407 | Firmicutes | Clostridia | Clostridiales | Ruminococcaceae | Ruminococcaceae_UCG-010 | | | | ✓ | | | | | | |
| Zotu247 | Firmicutes | Clostridia | Clostridiales | Ruminococcaceae | Ruminococcaceae_UCG-014 | | | | ✓ | | ✓ | | ✓ | | |
| Zotu300 | Firmicutes | Clostridia | Clostridiales | Ruminococcaceae | Ruminococcaceae_UCG-014 | | | | ✓ | | | | | | |
| Zotu199 | Firmicutes | Clostridia | Clostridiales | Ruminococcaceae | Ruminococcaceae_UCG-014 | | | | ✓ | | | | | | |
| Zotu373 | Firmicutes | Clostridia | Clostridiales | Ruminococcaceae | Ruminococcaceae_UCG-014 | | | | ✓ | | | | | | |
| Zotu193 | Firmicutes | Clostridia | Clostridiales | Ruminococcaceae | Ruminococcus_1 | | | | ✓ | | | | ✓ | | ✓ |
| Zotu394 | Firmicutes | Clostridia | Clostridiales | Ruminococcaceae | Ruminococcus_1 | | | | ✓ | | | | ✓ | | ✓ |
| Zotu158 | Firmicutes | Clostridia | Clostridiales | Ruminococcaceae | Subdoligranulum | | | | ✓ | | ✓ | | | | |
| Zotu375 | Firmicutes | Erysipelotrichia | Erysipelotrichales | Erysipelotrichaceae | Erysipelatoclostridium | ramosum | | | ✓ | | | | | | |
| Zotu54 | Firmicutes | Erysipelotrichia | Erysipelotrichales | Erysipelotrichaceae | Erysipelotrichaceae_UCG-003 | | ✓ | | ✓ | | ✓ | | ✓ | | |
| Zotu83 | Firmicutes | Erysipelotrichia | Erysipelotrichales | Erysipelotrichaceae | Holdemanella | biformis | | | ✓ | | ✓ | | ✓ | | |
| Zotu40 | Firmicutes | Erysipelotrichia | Erysipelotrichales | Erysipelotrichaceae | Holdemanella | | | | ✓ | | ✓ | | ✓ | | |
| Zotu47 | Firmicutes | Negativicutes | Selenomonadales | Veillonellaceae | Dialister | invisus | | | ✓ | | ✓ | | ✓ | | ✓ |
| Zotu323 | Proteobacteria | Alphaproteobacteria | Rhodospirillales | | | | | | ✓ | | ✓ | | ✓ | | ✓ |
| Zotu283 | Proteobacteria | Alphaproteobacteria | Rhodospirillales | | | | | | ✓ | | | | | | |
| Zotu261 | Proteobacteria | Alphaproteobacteria | Rhodospirillales | | | | | | ✓ | | | | | | |
| Zotu183 | Proteobacteria | Deltaproteobacteria | Desulfovibrionales | Desulfovibrionaceae | Desulfovibrio | | | | ✓ | | ✓ | | ✓ | | |
| Zotu255 | Proteobacteria | Deltaproteobacteria | Desulfovibrionales | Desulfovibrionaceae | Desulfovibrio | | | | ✓ | | ✓ | | ✓ | | |
| Zotu116 | Proteobacteria | Gammaproteobacteria | Betaproteobacteriales | Burkholderiaceae | Sutterella | | | | ✓ | | ✓ | | ✓ | | |
| Zotu307 | Proteobacteria | Gammaproteobacteria | Betaproteobacteriales | Burkholderiaceae | Sutterella | wadsworthensis | | | ✓ | | ✓ | | ✓ | | |
| Zotu53 | Proteobacteria | Gammaproteobacteria | Enterobacteriales | Enterobacteriaceae | Escherichia/Shigella | | ✓ | | ✓ | | | | | | |
| Zotu369 | Tenericutes | Mollicutes | Anaeroplasmatales | Anaeroplasmataceae | Anaeroplasma | | | | ✓ | | | | | | |

**Fig. 2 | Selection of ASVs (rows) that were significantly associated with depressive symptom levels in unadjusted analyses (Model 0) and results of subsequent adjusted analyses (Models 1–3).** Bars indicate effect size (standardized regression coefficient). Green bars indicating a positive and Red bars indicating a negative association (plotted range $0.10 \geq \beta \geq -0.10$). Checkmark indicate $p < 0.05$. The column "Core" highlights ASVs with >75% overall prevalence in the sample population (indicated by green check mark).

## Associations are mostly invariant across ethnic groups

Applying each of the 3 regression models, only a small proportion of ASV's (<6%) exhibited a significant ethnicity by ASV interaction (unadjusted for multiple testing), which thus approximated an expected Type 1 error rate. Ethnicity-stratified analysis of the age and gender-adjusted associations showed that most standardized regression coefficients (81%; $N = 337$) had a $I^2$ below 30% and only 15 correlations (3.6% of total) showed a substantial ethnic heterogeneity ($I^2 > 50\%$) (see Supplementary Data files 4, 5 and corresponding source data Supplementary data file 3).

## Core microbiota similarly associate with depressive symptoms

It is proposed that 'core taxa', i.e., bacteria with a near ubiquitous presence, may exhibit a stronger relevance to health[25]. Hence, auxiliary analyses compared the results obtained for all ASV to the results obtained for a core subset of taxonomic units (defined as ASVs with a ≥75% prevalence across ethnic groups). Because these highly prevalent core taxa are minimally zero-inflated, these comparisons additionally function as sensitivity analyses for zero-inflation bias. Comparisons between core and non-core taxa revealed no differences with regard to the proportion of significant associations with depression, the average or distribution of effect sizes, robustness to covariate adjustments, or ethnic heterogeneity ($I^2$) in the associations with depression.

## Discussion

A primary aim of the current study was to identify which taxonomic features of the gut microbiota are linked to depressive symptom levels. This investigation involved the largest study cohort to date examining microbiome-depression associations, and is the first to study ethnicity as a potentially relevant factor in this association. Consistent associations between the gut microbiota and depressive symptom levels were confirmed at multiple levels of analysis, ranging from global parameters of microbiota diversity (i.e., α-diversity, β-diversity) to the relative abundances of specific taxa. These associations withstood adjustment for a broad range of sociodemographic, behavioral, and medical covariates. Analyses further revealed that these associations were largely invariant across ethnic groups, notwithstanding the substantial ethnic differences in both depressive symptom levels and composition of the gut microbiota[14,18,19]. Moreover, ethnic disparities in depressive symptom levels were partly explained by between-subject differences in microbiota composition (i.e., β-diversity)[19].

Inspection of (mutually adjusted) regression coefficients revealed α-diversity predicted depressive symptoms with effect sizes comparable to several other established risk factors of depression, such as alcohol consumption, exercise, smoking, and BMI[26]. Conversely, the ability of depressive symptoms to statistically predict α-diversity was in the same range as being diagnosed with, for example, diabetes or a GI disorder[11–16]. By implication, then, these analyses suggest that conditions and interventions that influence the gut microbiome may have the potential to impact well-being on a population-level.

A notable finding was that both Bray-Curtis and weighted UniFrac Principal Component #2 shared substantial variance with α-diversity (Shannon). This finding indicates that α-diversity (a measure of within-subject microbial diversity) also meaningfully characterized between-subject diversity (beta-diversity); in other words, taxa that correlate highly with α-diversity are unevenly distributed across individuals. This is a pertinent observation because exactly these taxa also tended to correlate with depressive symptom scores, as well as established risk factors of depression (e.g., BMI, inflammation, diabetes)[27–29]. The latter

replicates prior findings[12,30–32]. Taken together, then, these results are consistent with the idea of α-diversity as a generic biomarker of health and vulnerability[33,34] (including depression), as well as with the notion of a common set of bacteria that tend to non-specifically respond to disease and poor health[35].

The association of α-diversity with depression dissolved after adjustment for the personality trait neuroticism, which is a constitutional and generic risk factor for common mental disorders, including depression[22]. This dominant effect of neuroticism might help clarify the observation that disruptions in the gut microbiome have been associated with a rather broad range of psychological disorders without concomitant evidence of taxonomic specificity (i.e., whereby specific taxa differentiate specific disorders)[4,8,36]. Of note, the other principal components of β-diversity appeared impervious to adjustment by neuroticism, and these might thus identify the more depression-specific features of microbiota composition.

Unadjusted analyses of relative abundances initially yielded 117 ASVs (identifying 59 genera, mostly belonging to the phylum *Firmicutes*) that correlated with depressive symptom scores. Significantly, many of those taxa have also been linked to other domains of health[32,35,37–41], including health factors associated with increased depression risk (e.g., BMI). A prominent example was the genus *Christensenellaceae* (*R-7 group*), which likewise showed a negative association with depressive symptoms in the Rotterdam study cohort[24]. Further, and replicating prior research, *Christensenellaceae* abundances were additionally correlated with lower BMI and relatively depleted in the presence of diabetes and gastro-intestinal diseases[42]. These links with medical outcomes may therefore explain why most associations with *Christensenellaceae* became nonsignificant after adjustment for the medical covariates.

After full adjustment (i.e., Model 3), 23 ASVs identifying at least 15 genera remained significantly associated with depression scores. These included a negative association with the abundant genus *Coprococcus* (designated GCA-900066575 in the Silva database), hereby confirming results from two independent population cohorts[13] and the Rotterdam study[24]. This genus harbors many butyrate-producing species and has been ascribed anti-inflammatory properties[43], both of which have been (inversely) linked to depression. Our analyses showed a positive associations with the genus *Dialister*[13], which ASV we could map onto the oral pathogen *D. invisus*[44]. This observation is in step with data showing that poor oral health is a correlate of depression[45,46] and a possible upstream determinant of the gut microbiota[47].

Other bacteria relatively depleted in relation to depressive symptoms were the Bacteriodetes genus *Bacteroides*, *Ruminococcaea* UCG005, *Ruminococcus 1*, *Peptococcus*, *Holdemanella* (sp. *H. biformis*), various genera in the family *Lachnospiraceae*, e.g., *Lachnospiraceae* groups FCS020 and NK4A136, *Marvinbryantia* (sp. *M. formatexigens*), *Blautia* (among which the species *B. obeum*, which was until recently classified under the genus *Ruminicoccus* and exhibits overlapping physiological characteristics[48]), *Roseburia* (sp. *R. inulinivorans*), and the Proteobacteria genus *Desulfovibrio*. Simultaneously, five ASVs were enriched in those with high symptom levels. These included the *Blautia* species *B. caecimuris* and *B. producta*, the genera *Lachnoclostridium* and *Oscilibacter*, and the aforementioned *Dialister invisus*. Overall, a majority of associations were within the phylum Firmicutes. Random forest analyses, using the Rotterdam Study as training cohort and HELIUS as the testing cohort[24], replicated associations for several including R*uminococcaceae UCG005*, *Coprococcus*, *Lachnoclostridium*,

*Eggerthella, Sellimonas, Roseburia, Bacteroides, Blautia, Veillonella*, and *Desulfovibrio*, of which several were also retained in their adjusted analyses[24].

Unexpectedly, ASVs identifying the genus *Bifidobacterium* showed a positive correlation with depression[36], including the abundant species *B. longum* (greengenes database) that has been tested in multiple probiotic studies for its potential to enhance mood[49–51]. In parallel, we found that *B. longum* was negatively associated with α-diversity, which replicates observations from another large cohort[12], while simultaneously showing the (expected) negative associations with some markers of poor health[32]. The *Bifidobacterium* genus is highly diverse[52] and it therefore conceivable that the previously reported beneficial mood effects of supplementation are highly strain-specific. Probiotic trials using this genus have yielded inconsistent results whereby a majority have failed to establish beneficial mood effects[49–51]. The results presented here may help identify other candidates for psychobiotic interventions.

The present results suggested that the custom of analysing bacteria at an aggregate level (e.g., genus, OTU) may be a potential cause of inconsistent findings in microbiome-depression studies[53]. It may thus also clarify some variance with the study of Radjabzadeh et al.[24], which used closed reference OTU clustering instead of ASVs. For example, we observed that ASVs identifying the genus *Blautia, Bacteroides*, or *Oscillospira* exhibited both significant positive and significant negative correlations with depressive symptom levels. Such potentially biologically meaningful associations may average out when aggregated on a genus level. For example, heterogeneous associations within the same genus may reflect interspecific competition as ecological competition is known to be especially fierce within the same genus[54], i.e., indicating that depression is associated with a gut environment conducive to some species while disadvantageous to others[55,56]. Imprecisions in taxonomic allocation could likewise account for heterogeneous associations[57,58]. For example, the notably heterogeneous pattern of associations between depression and 10 ASVs that Greengenes maps onto the genus *Oscillospira* disappeared when utilizing the Silva database (which assigned these 10 ASVs to 8 different genera). Together these observations align with the burgeoning view that ASVs are preferred as the standard unit of marker-gene analysis and reporting[53,59].

In closing several strengths and limitations warrant mentioning. The present analyses pertained to ethnic groups living in the same urban area, hereby preventing confounding by geographical effects[15]. While ethnic differences in the gut microbiota may involve both genetic and environmental factors, the balance of evidence seems to indicate that the latter may dominate[60–62]. More fine-grained analyses, e.g., comparing 1st with 2nd generation immigrants, or comparing the history of local acculturation among 1st generation migrants, may further identify the specific role of environmental exposures. Another advancement is that the present study applied an unprecedented confounder adjustment. Future studies may still consider additional explanatory factors (e.g., diet[63,64]). We may add that the attenuation of effect sizes with progressive covariate adjustments should not be taken as indicative of spurious associations, since some covariates may be on the causal pathway[65]. We note that rating instruments like the PHQ-9 do not provide a clinical diagnosis of depression, although these assessment approaches tend to be highly correlated[66]. The use of a continuous symptom-score is more in step with contemporary views of depression as a continuum[67]. Depression is a heterogenous construct with different subtypes based on symptom profile. This aspect warrants further research in light of evidence that such subtypes may exhibit distinct biological profiles[68,69].

Three additional statistical considerations warrant mentioning also: a strength is that key analyses were cross-validated using different methods, e.g., utilizing multiple parameters of alpha-diversity and beta-diversity, the comparative use of two data bases for taxonomic

allocation, and performing both meta-analysis and GLM to determine ethnic heterogeneity. The fact that these different approaches yielded a comparable pattern of results supported the robustness of the present findings. Further, in the context of p-value testing and small effect sizes, the observation that one study or subsample shows a significant association and the other does not, cannot immediately be taken as evidence of a non-replication or inconsistency (e.g., see Radjabzadeh et al.[24] as well as Supplementary Fig. 1 in the current paper), but may reflect normal between-sample variation and Type 2 error. Large-scale aggregate analyses of multiple cohorts may be a relevant approach therefore. Finally, although depression symptoms were modeled as the outcome variable in most analyses, causal inferences obviously remain speculative at this point.

In summary, analyses of a large and ethnically diverse population demonstrated robust associations between the gut microbiota and depressive symptoms. These associations were largely invariant across ethnic groups and withstood adjustment for a uniquely large set of relevant confounders, which included demographic, behavioral, and medical factors. The study findings identified potential targets for psychobiotic interventions that warrant further investigation, and may positively impact depression and well-being at an individual or population level.

## Methods

### Procedures and participants

The HELIUS (Healthy Life in an Urban Setting) study is a multi-ethnic cohort study among citizens of Amsterdam, The Netherlands[20,21]. The city proper is a moderately sized area (219.49 km²) with approximately 900,000 inhabitants, and is the national capitol. The full study protocol is described elsewhere[20,21]. In short, participants aged 18–70 years were randomly sampled, stratified by ethnic origin, through the municipal registry of Amsterdam (participation $N = 24,789$, response rate 28%). Data were collected through physical examination and by questionnaire, which was either self-administered or collected by interview using an ethnically matched interviewer. The HELIUS study was complied with all relevant ethical regulations and in accordance with the Declaration of Helsinki (6th, 7th revisions). Written informed consent was obtained from all participants prior to inclusion. The study was approved by the Institutional Review Board of the Amsterdam University Medical Centers, location AMC.

At the time of the present analyses, fecal 16 S rRNA data were available for a total of 3.343 participants belonging to 8 ethnic groups. Because of small numbers, those identifying as Indonesian-Surinamese background ($N = 46$) and "another or unknown ethnicity" ($N = 63$) were excluded. Applying these criteria, and excluding those without data on depressive symptoms (PHQ-9, see below; $N = 93$), yielded the following 6 ethnic groups; Dutch ($N = 769$), African Surinamese ($N = 767$), South-Asian Surinamese ($N = 527$), Turkish ($N = 349$), Moroccan ($N = 473$), and Ghanaian ($N = 458$). Ethnic groups were classified on the basis of migratory background[14,20,21]. Accordingly, a person was considered to be of non-Dutch ethnicity when meeting one of the following two criteria: (1) born outside the Netherlands and at least one parent born outside the Netherlands (i.e., first generation), or; (2) born in the Netherlands with both parents born outside the Netherlands (second generation). For participants with a Surinamese ethnicity further subgroups were identified according to self-described ethnic origin[14,20]. For the Dutch sample, we only invited people who were born in the Netherlands and whose parents were born in the Netherlands.

### Depressive symptoms, sociodemographic, behavioral, and medical variables

Depressive symptoms were recorded using the 9-item Patient Health Questionnaire-9 (PHQ-9)[66,70]. The PHQ-9 boasts good psychometric properties and has been shown to measure the same concept (i.e., is invariant) across all six ethnic groups included in this study[19,66,71]. Each

of the PHQ-9 items evaluates the presence of one of the nine DSM-IV symptom criteria experienced during the past 2 weeks, utilizing a four-point Likert-scale (not at all - almost every day). The severity of depressed mood was assessed by the sum score (ranging between 0 and 27). In case of a single missing item the mean score of the remaining items was used to replace the missing item; with >1 missing items the entire PHQ-9 was considered missing[18].

Data on sociodemographic, behavioral, and medical variables were collected by self-report or physical examination[20]. Demographic data included ethnicity, sex, age, educational level. The latter comprised 4 categories ranging between 'elementary education or less' and 'advanced vocational or university education' (i.e., BA/BSc or higher). Analyses of behavioral factors focused on physical activity (i.e., minutes per week times the intensity activity of each minute-activity; as categorized in 3 METs groups as based on the compendium of Ainsworth[72]), smoking (yes/no), alcohol (alcohol consumption and alcohol-related problem behaviors, as assessed by the 10-item Alcohol Use Disorders Identification Test[73]), and body mass index (BMI). Medical covariates included a (self-reported) diagnosis of a gastro-intestinal disorder and diabetes. The latter was established using a Boolean algorithm whereby caseness was classified as a self-reported clinical diagnosis, or increased fasting glucose (≥7 mmol/l), or increased HbA1c (≥48 mmol), or the use of glucose-lowering medication. Medication and supplement intake were also recorded and included the use of proton pump inhibitors, antidepressants, and use of antibiotics (past 2 weeks), as well as use of probiotics. Participants also reported symptoms of diarrhea experienced over the past week. Data on inflammatory activity (plasma C-reactive protein) were available for a subset of participants ($N = 975$) and only used in auxiliary analyses. Approximately 26% of questionnaires was filled out with support of an interviewer-assistant at the assessment center. This entailed that an ethnically and language-matched person was present at the assessment center to provide clarification due to language or reading problems. These assistants were trained to ensure standardization. Follow-up studies yielded no response differences by either administration modus (paper-pencil, digital, assistant supported) or ethnicity on the main outcome variable (GHQ-9)[19]. Further, analyses showed minimal ethnic and socio-economic differences between participants and those who declined or did not respond[21].

## Stool sample collection

Participants were given a stool collection tube and a safety bag (for transport) either through mail before assessment center visit or at the end of the visit, as preferred. They were asked to bring a 'fresh' stool sample to the assessment center within 6 h after collection. If not possible, participants were instructed to keep the stool sample in their home freezer overnight and to bring it in frozen to the research location the next morning. All samples were immediately frozen at −20 °C at each assessment center, and transported within 1 to 4 weeks to the University medical Center and frozen at −80 °C until processing. The time period each sample was stored locally at −20 °C was not logged. During the physical examination, asked if (1) they used probiotics (frequency, type), (2) used antibiotics in the past three months or two weeks, (3) had experienced diarrhea in the past week. Standardized procedures were used at the collection sites using written SOPs and training of research personnel. Quality checks on the staff/procedures were done at regular intervals during the data collection period.

## Bioinformatics

Fecal microbiota composition was profiled by sequencing the V4 region of the 16S rRNA gene on an Illumina MiSeq instrument (Ilumina RTA v1.17.28; MCS v2.5) with 515F and 806R primers designed for dual indexing[74] and the V2 Illumina kit (2 × 250 bp paired-end reads)[14]. 16S rRNA genes from each sample were amplified in duplicate reactions in

volumes of 25 µl containing 1x Five Prime Hot Master Mix (5 PRIME GmbH), 200 nM of each primer, 0.4 mg/ml BSA, 5% DMSO, and 20 ng of genomic DNA. PCR was carried out under the following conditions: initial denaturation for 3 min at 94 °C, followed by 25 cycles of denaturation for 45 s at 94 °C, annealing for 60 s at 52 °C and elongation for 90 s at 72 °C, and a final elongation step for 10 min at 72 °C. Duplicates were combined, purified with the NucleoSpin Gel and PCR Clean-up kit (Macherey-Nagel) and quantified using the Quant-iT PicoGreen dsDNA kit (Invitrogen). Purified PCR products were diluted to 10 ng/µl and pooled in equal amounts. The pooled amplicons were purified again using Ampure magnetic purification beads (Agencourt) to remove short amplification products. Raw sequencing reads were quality checked using FastQC. USEARCH (v11.0.667 64-bit Linux version)[75] was used to process the raw reads. Read pairs were merged with 30 maximum accepted differences and 80% minimum overlap identity, then filtered using a threshold of maximum 1 expected error per merged contig. Reads passing the filter were subsequently dereplicated. Sequences occurring at least 8 times in the entire dataset were used to infer biological sequences with the UNOISE3 algorithm (α-parameter set to 2.0)[76]. All merged reads (including reads that failed quality filtering) were mapped back to the inferred Amplicon Sequence Variants (ASVs) in order to construct an ASVs table[59]. Taxonomy was assigned to the ASVs with the SINTAX algorithm[77] using Greengenes v.13.5 and Silva 132[78]. The ASV table was rarefied to 14,942 counts per sample. ASV sequences were then used as input for MAFFT (v.7.427)[79,80] in order to obtain a multiple sequence alignment, based on which a phylogenetic tree was constructed using IQ-TREE (v. 1.6.11)[48]. The phylogenetic tree was midpoint-rooted using the "phytools" R package (Revell, 2012). The "phyloseq" R package[81] was used to integrate the ASV counts, taxonomy assignments, phylogenetic tree and sample metadata. The above analyses identified 1438 ASVs of which 418 were deemed to have a non-trivial counts (>0.02%, corresponding to approximately 3 counts per sample out of 14,942 reads). A core microbiota subset was defined on the basis of ASVs that were present in at least 75% of the cohort, yielding 109 ASVs.

## Covariate selection

To avoid overfitting, covariates were selected a priori, as informed by prior epidemiological analyses (mainly[11,12,82] and insofar available in our dataset). The selected covariates involved sociodemographics (ethnicity, age, sex, education), behavioral/lifestyle (smoking, alcohol, exercise, bodyweight (BMI), and medical variables (diabetes, diagnosis of GI disorder, proton-pump inhibitor (PPI) use, recent antibiotic use, recent diarrhea; see "Statistical methods" section). Never-smokers and former smokers were categorized as non-smokers. Post hoc analyses yielded comparable results when former smokers were omitted from analyses (results not shown here). To ascertain that no important medical covariates were overlooked, exploratory analyses were performed to identify variables with additional explanatory value in the fully adjusted models. Among the variables tested were presence of metabolic syndrome and its components[83], glucocorticoid medication, statins, beta-blockers, inflammatory diseases. None of these were retained in the final analyses on the basis of failing to significantly alter the association between predictor and main outcome. In follow-up of recent recommendations[13], the small number of antidepressant users were excluded from the main analyses ($N = 132$). Auxiliary analyses included covariates that reflect pre-existing psychological risk factors for depression, for which we selected parental history of depressive disorders, number of prior depressive episodes, and neuroticism.

## Statistical methods

Statistical analyses were performed in R version 4.0.1, SPSS v27, or JASP 0.13.1. Multiple linear regression was used to determine the association between α-diversity or β-diversity and depressed mood (PHQ-9 sum scores), with the latter as the outcome variable. Covariates (see above)

were added in a stepwise fashion yielding 3 models; Model 1 was adjusted for sociodemographics (age, sex, ethnicity, education), Model 2 additionally adjusted for health-related behaviors (smoking, alcohol, exercise, BMI), Model 3 further incorporated medical variables (diabetes, self-reported diagnosis of GI disorder, use of proton pump inhibitors, antibiotic use past 2 weeks, diarrhea past week).

Alpha- and Beta-diversity indexes were calculated using the 'vegan' package in R[84] (see above). The Shannon index was used as the primary marker of α-diversity, but analyses were repeated for other measures of α-diversity (i.e., Phylogenetic Diversity, Chao1, Abundance-based Coverage Estimator Observed, Simpson index). For β-diversity, principal coordinate analyses (PCoA) were performed using weighted UniFrac metrics and Bray-Curtis distances. The first 20 principal coordinates were selected for inclusion in multivariable regression analyses by applying forward selection, and the resulting coordinates were used as predictor variables in linear regression (see further description in the results section), utilizing the same 3 regression models described above. The three regression models were also used to examine association between individual ASVs and depressed mood. For these analyses both the predictor (PHQ-9 sum scores) and independent variables (relative abundances) were rank-ordered to yield a more robust estimate. $P$-values were FDR-corrected for multiple testing (Benjamini-Hochberg)[85]; a corrected $P$-value < 0.05 was considered statistically significant.

Possible heterogeneous between-ethnic associations were examined by two methods: First, GLM (SPSS UNIANOVA) was used to determine significant interactions between ethnicity and each ASV (FDR adjusted[85]). As a second method, the associations were stratified by ethnicity and the heterogeneity of microbiota-depression associations was quantified as $I^2$ (i.e., comparable to a meta-analysis). Associations showing a $I^2$ > 30% and >50% across ethnicity are considered to reflect moderately or high heterogeneity, respectively[86].

Mediation analyses were used to test if beta-diversity (i.e., microbial diversity between individuals) may statistically account for ethnic disparities in depressive symptom levels, following the inferential steps as described by Kenny and Baron[87].

### Reporting summary
Further information on research design is available in the Nature Research Reporting Summary linked to this article.

## Data availability
Raw sequence data (Illumina MiSeq 16S rRNA sequencing, V4 hypervariable region) can be found at https://ega-archive.org/datasets/EGAD00001004106. The HELIUS data are owned by the Amsterdam University Medical Centers, located at the AMC in Amsterdam, The Netherlands. Any researcher can request the data by submitting a proposal to the HELIUS Executive Board as outlined at http://www.heliusstudy.nl/en/researchers/collaboration, accessed on 28 March 2022, by email: heliuscoordinator@amsterdamumc.nl. The HELIUS Executive Board will check proposals for compatibility with the general objectives, ethical approval and informed consent forms of the HELIUS study. There are no other restrictions to obtaining the data and all data requests will be processed in the same manner.

## Code availability
Software description and syntax of analyses can be found at https://amcmc.github.io/HELIUS_depression/.

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

## Acknowledgements

The HELIUS study is conducted by the Amsterdam University Medical Centers, location AMC and the Public Health Service of Amsterdam. Both organizations provided core support for HELIUS. The HELIUS study is also funded by the Dutch Heart Foundation, the Netherlands Organization for Health Research and Development (ZonMw), the European Union (FP-7), and the European Fund for the Integration of non-EU immigrants (EIF). We are most grateful to the participants of the HELIUS study and the management team, research nurses, interviewers, research assistants and other staff who have taken part in gathering the data of this study. We especially like to thank Dr. Henrike Galenkamp, Dr. Mary Nicolai, Prof. Aart Schene, Dr. Laura Steenbergen, Prof. Karien Stronks, and Dr. Andrei Prodan for their support in various phases of the project. Grant numbers: - Dutch Heart Foundation: 2010T084 (K Stronks). - ZonMw: 200500003 (K Stronks). - European Union (FP-7): 278901 (K Stronks). - European Fund for the Integration of non-EU immigrants (EIF): 2013EIF013 (K Stronks). - H2020 Research Innovation Action (RIA). Grant agreement ID: 848146 (JA Bosch).

## Author contributions

J.B. drafted the work and substantively revised it. J.B., M.N., A.Z., M.D., D.R., R.K., M.D., S.R., and A.L. all had substantial contributions to the conception or design of the work, or the acquisition, analysis, or interpretation of data. All authors have approved the submitted version and have agreed to be personally accountable for the author's own contributions and to ensure the accuracy or integrity of any part of the work.

## Competing interests

M. Nieuwdorp is founder and a member of the Scientific Advisory Board of Caelus Health, The Netherlands. However, none of these conflicts bear any relevance to the content of the current paper. The other authors declare no conflicts of interest. The funders had no role in the design of the study, in the collection, analyses, or interpretation of data, in the writing of the manuscript, or in the decision to publish the results.
