## [Peer Review File · Nature Communications]

REVIEWER COMMENTS

Reviewer #1 (Remarks to the Author):

The current study investigated associations between gut microbiota and depressive symptoms in a large population-based cohort. In the analyses they adjusted for possible confounders including demographics, lifestyle and medical factors. Furthermore, they compared the associations across different ethnic groups. The authors found that the association between the gut microbiota and depressive symptoms was robust across ethnicity and several confounding factors.

In general, the study is well performed, and the manuscript is clear and well written. I commend the authors for a nice piece of work.

I, however, have a few comments:

Major comments:

Methods

- Did the questionnaire take recent smokers into account – i.e. those that stopped smoking prior to the study, but who had been heavy smokers before that?
- How many of the samples were initially frozen at -20 °C before transfer to -80 °C and how many of the samples were immediately frozen at -80 °C without the intermittent storage at -20 °C. And were the differences evenly spread out between ethnic groups?
- Details on who provided self-administered physical exams and questionnaires, and who provided them through an interview is necessary. Was this evenly spread out through the ethnic groups? Could there have been imposed biases here?

Table 1:

- Gender and age should start with capital letters
- Specify in the table text what the “M” stands for.

-

- It appears as if the mean for moderate physical activity is 2592 minutes / week per participant? Which is probably not true. It should be clarified that whether it is the total for the group, or the individual physical activity.

Table 3:

- In the PDF version, the first row has been cut, and thus not all the text can be seen. I would recommend increasing the row height of this first row.

Minor comments:

- L. 90. Be advised, that the term "microbiota" is preferred over "microflora".

- When specifically referring to the bacteria, and not their genetics or metabolome, the term "microbiota" is often preferred over "microbiome".

- L. 247-268. The authors shift between whether they are writing the first "0" when writing decimals. E.g. for the overall model fit, the authors write " $\Delta R^2 = 0.0597$, $p < .001$ " while when lifestyle is added, they write " $\Delta R^2 = .0087$, $p < .001$ ". While both is fine for me, I would prefer if the same principal were used throughout the text.

Reviewer #2 (Remarks to the Author):

My hypothesis would be that the connection between the microbiome and ethnicity would be difficult to disentangle due to factors such as diet, similar practices (e.g., around pets, activities such as being out in nature), similar experiences (e.g., racism). Although you cite Deschasaux et al, saying it showed ethnicity drove microbial differences in OTUs and alpha diversity that was not related to lifestyle, diet. I think you need to make a better case for this in this article. You should probably mention studies such as Rothschild's "Environment dominates over genetics in shaping human gut microbiota". There is support for genetic components with Goodrich's "Genetic Determinants of the Gut Microbiome in UK Twins." I found you have an article from HELIUS from 2017 about dietary acculturation among the South-Asian Surinamese population in the Netherlands PMID: 27122356.

Could you look at first generation versus second generation? Or did that make the numbers too small in each group? Did you account for how long ago the subject or the subject's family immigrated? Is there a way to account for acculturation stress?

Do you need to consider factors such as trauma?

Can you further define living in the same urban area? How large is that urban area?

You also need to present more background on your prior work with differences in depression among these groups to understand your hypothesis better. You note that depression is heterogenous but then do not really look at it beyond the PHQ-9.

In the Intro you bring up that the literature has sparse adjustments for factors such as lifestyle. you have attempted to include factors such as exercise (although how accurate is minutes per week), smoking, alcohol use but does not really account for diet (beyond metabolic factors such as HbA1C.

A small thing, but be consistent throughout with things like PPI versus proton pump inhibitor--PPI used before proton pump inhibitor spelled out.

For methods, the first twenty principal coordinates were used--often this many are not used. What was the rationale?

My understanding is that Greengenes has not been updated in multiple years and this is a limitation of it.

Your conclusion is not clear. You state the associations are largely invariant but that the microbiota may partly account for depression disparities. Neuroticism should have been brought up sooner as you say alpha-diversity no longer differed when also including neuroticism. What is the take-home message? What are next steps?

POINT BY POINT REPLY TO THE REVIEWER COMMENTS

Reviewer #1 (Remarks to the Author):

In general, the study is well performed, and the manuscript is clear and well written. I commend the authors for a nice piece of work.

We thank the reviewer for the kind words and the regard for our work. We are hopeful that the replies below will help reaffirm this assessment.

Major comments:

Methods

- Did the questionnaire take recent smokers into account – i.e. those that stopped smoking prior to the study, but who had been heavy smokers before that?

For the current analyses, we have lumped non-smokers and former smokers into a single category. We agree that this approach may possibly introduce some noise/measurement inaccuracy. To assess this possibility, we re-ran our analyses omitting all former smokers (n=1440; 25.6%).

The results showed that for alpha-diversity (i.e., Shannon predicting PHQ-9) the standardized regression coefficients came out virtually identical or marginally stronger, with comparable p-values: for Model 2 standardized Beta was .060 vs .068; Model 3 standardized Beta was .042 vs .048. Extending these re-analyses to abundances likewise yielded a comparable pattern of results.

It is conceivable that more fine-grained analyses (e.g., focusing on recent quitters and especially former heavy-smokers) might reveal that those who stopped recently may continue to exhibit smoker-like effects for a prolonged period, but we felt that such analyses would fall outside the scope of the current paper. The analyses seemed to indicate that the effects of possible measurement imprecision on the results as reported here are modest.

The above analyses have now been noted in the methods section on page 8:

“Never-smokers and former smokers were categorized as non-smokers. Post-hoc analyses yielded virtually identical results when former smokers were omitted (results not shown here).”

How many of the samples were initially frozen at -20 °C before transfer to -80 °C and how many of the samples were immediately frozen at -80 °C without the intermittent storage at -20 °C. And were the differences evenly spread out between ethnic groups?

All samples were immediately frozen at -20C at the research location, and so there are no differences between research locations (or ethnic groups) in this aspect. The time until transportation to the Academic Medical Center varied between weekly and once every 4 weeks (whenever the freezer was filled-up). Immediately upon arrival at the AMC the samples were frozen at -80. There is no registration of how long each batch was stored at -20C before transportation.

This information has now been added to the methods, on page 7

“All samples were immediately frozen at -20C at each assessment center, and transported within 1 to 4 weeks to the University medical Center and frozen at -80C until processing. The time period each sample was stored locally at -20C was not logged.”

Details on who provided self-administered physical exams and questionnaires, and who provided them through an interview is necessary. Was this evenly spread out through the ethnic groups? Could there have been imposed biases here?

Approximately 26% of questionnaires was filled out with support of an interviewer/assistant at the assessment center. This entailed that an ethnically and language-matched person was present at the data collection site to address any questions (e.g., provide clarification when requested , e.g., due to language or reading problems). These assistants were trained to ensure standardisation.

A possible ethnic or method (paper-pencil, digital, assistant supported) bias is conceivable, and we have therefore tested if ethnicity and modus of administration may have caused measurement variance of the PHQ-9, our primary outcome variable, and other questionnaires (sf-12) (results published in; Galenkamp H. et al. 2018 Plos One; Galenkamp H. et al. 2019, BMC Psychiatry). These studies yielded no response differences by either administration modus or ethnicity. We felt that these findings provided confidence in the validity/representativeness of our results. We may add that education was included in all regression models, which hereby also adjusted for literacy levels. Further, non-response analyses show that ethnic and socio-economic differences between participants and nonparticipants (i.e., those who declined the invitation or did not respond) were very small (Snijder MB et al. 2017, BMJ Open).

The above information has now been added to the methods section, on page 7, as follows:

“Approximately 26% of questionnaires was filled out with support of an interviewer-assistant at the assessment center. This entailed that an ethnically and language-matched person was present at the assessment center to provide clarification due to language or reading problems. These assistants were trained to ensure standardisation. Follow-up studies yielded no response differences by either administration modus (paper-pencil, digital, assistant supported) or ethnicity (Galenkamp H. et al. 2018 Plos One; Galenkamp H. et al. 2019, BMC Psychiatry). Further, analyses showed minimal ethnic and socio-economic differences between participants and those who declined the invitation or did not respond (Snijder MB et al. 2017, BMJ Open).”

Table 1:

- Gender and age should start with capital letters

Thanks, this is now fixed for both Table 1 and Table 2

- Specify in the table text what the “M” stands for.

M is now replaced with Mean.

- It appears as if the mean for moderate physical activity is 2592 minutes / week per participant? Which is probably not true. It should be clarified that whether it is the total for the group, or the individual physical activity.

We thanks the reviewer for careful reading. The text “by intensity” was accidentally missing; i.e., the number represents the number of minutes per week times the intensity activity of each minute-activity (categorized in 3 METs groups as based on the compendium of Ainsworth) (Wendel-Vos WGC, 2003, J Clin Epidemiol).

The Table 1 legend, Table 2, and relevant text in the methods (page 7) have now been adjusted accordingly

The text on page 7 now reads 7:

“Analyses of behavioral factors focused on physical activity (i.e., minutes per week times the intensity activity of each minute-activity; as categorized in 3 METs groups as based on the compendium of Ainsworth), ... etc.”

Table 3:

- In the PDF version, the first row has been cut, and thus not all the text can be seen. I would recommend increasing the row height of this first row.

Thanks. This has now been fixed.

Minor comments:

- L. 90. Be advised, that the term “microbiota” is preferred over “microflora”.

We agree, and this has been adapted through-out.

- When specifically referring to the bacteria, and not their genetics or metabolome, the term “microbiota” is often preferred over “microbiome”.

We again agree. We now use microbiota when pertaining to our own analyses and findings. On a several occasions we still use the term ‘microbiome’ when, in context, it is used to more broadly refer to the area of research.

- L. 247-268. The authors shift between whether they are writing the first “0” when writing decimals. E.g. for the overall model fit, the authors write “ $\Delta R^2 = 0.0597$, $p < .001$ ” while when lifestyle is added, they write “ $\Delta R^2 = .0087$, $p < .001$ ”. While both is fine for me, I would prefer if the same principal were used throughout the text.

Thanks again. This has been corrected (omitting the 0) and results are now reported in a consistent format, throughout the manuscript text and in Table 2.

Reviewer #2 (Remarks to the Author):

My hypothesis would be that the connection between the microbiome and ethnicity would be difficult to disentangle due to factors such as diet, similar practices (e.g., around pets, activities such as being out in nature), similar experiences (e.g., racism).

We thank the reviewer for the careful review of our paper, also with regard to the other points raised below. The reviewer is accurate to emphasize that ethnicity is a multidimensional construct, encompassing genetic ancestry, lifestyle factors, living environments, and unique exposures (e.g., specific stressors). We believe that this fact does not muddle interpretation of the results: The main observation was that ethnicity, as complex and encompassing it may be, does not seem to appreciably impact the associations between depressive symptom levels and microbiota composition. We believe that this is a noteworthy finding, as one may intuitively have anticipated the opposite, e.g., reasoning that since ethnicity touches on multiple dimensions, as pointed out by the reviewer, prior findings in those with north-European ancestry may not generalize to those with other ethnic backgrounds.

The second novel observation was that microbiota composition appears to partly account for the well-established ethnic differences in depressive symptoms levels. This association remained after adjustment for factors like education, BMI, age, gender, physical activity, alcohol, smoking, and several medical covariates. As we have now established statistical mediation, subsequent research becomes justified to identify the exact pathways and mechanisms involved. This may well include some of the factors mentioned by the reviewer, or may involve other pathways. While we could certainly pick several possible candidates for further exploratory analyses, such was not within the aim of our study; we foremost aimed to establish if, and to what extent, the robust ethnic differences in microbiota composition may contribute towards the parallel disparity in depression. In the absence of a mechanistic model we preferred to minimize post-hoc analyses (and the potential risk of spurious findings), and opted for a slightly more conservative approach.

Further below the reviewer makes a related point on which we elaborate further.

Although you cite Deschasaux et al, saying it showed ethnicity drove microbial differences in OTUs and alpha diversity that was not related to lifestyle, diet. I think you need to make a better case for this in this article. You should probably mention studies such as Rothschild's "Environment dominates over genetics in shaping human gut microbiota". There is support for genetic components with Goodrich's "Genetic Determinants of the Gut Microbiome in UK Twins." I found you have an article from HELIUS from 2017 about dietary acculturation among the South-Asian Surinamese population in the Netherlands PMID: 27122356.

The point of the reviewer is well taken, in that the marked ethnic differences in the gut microbiota may involve both genetic and environmental determinants (as well as their interaction). The introduction nor our study aimed to specifically address that issue. In the introduction we note that currently the epidemiological data is limited to populations of north-European descent. Hence, we raised (and tried to answer) the question to what extent prior findings generalize to other ethnic populations, irrespective of what explains such ethnic differences. That is not to say, however, that the latter does not warrant further scrutiny, and we now elaborate on that topic in the discussion. On page 15:

“Ethnic differences in the gut microbiota may be accounted for by both genetic and environmental factors, whereby at current the balance of evidence seems to suggest that impact of the latter may dominate {Goodrich, 2016; Groot, 2020; Rothschild, 2018}. More fine-grained analyses, e.g., comparing 1st with 2nd generation immigrants, or the history of local acculturation among 1st generation migrants, may help identify relevant environmental exposures that explain ethnic differences.”

Could you look at first generation versus second generation? Or did that make the numbers too small in each group? Did you account for how long ago the subject or the subject's family immigrated? Is there a way to account for acculturation stress?

Separate analyses would indeed have low statistical power as approximately 10% of ethnic participants were 2nd generation migrants. However, in response to the query of the reviewer we did re-run our alpha and beta diversity analyses adding migration-generation as a covariate. The results came out near identical both in terms of p-values and effect sizes. However, the results of these post hoc analyses do certainly not exclude the (likely) possibility that comparisons between subsequent generations, or separate analyses of longer periods of local acculturation of 1st generation migrants, may show a diminution of some ethnic differences. Our findings should merely be taken to indicate that in the present sample such differential acculturation did not seem to impact the reported findings and conclusion. Overall we felt that further fine-grained within-ethnic analyses were outside the scope of the current paper, and we opted for a slightly more conservative approach to exploratory analyses. We now report this as a limitation in the discussion on page 15

“A more fine-grained analyses, e.g., comparing 1st and 2nd generation immigrants, or comparing the duration of local acculturation among 1st generation migrants, may help identify the role of specific environmental exposures.”

Do you need to consider factors such as trauma?

There are several possible risk factors for depression that may be unevenly distributed among ethnic groups. Psycho trauma might be one of those, but its possible mechanistic role in mediating the depression-effects on the gut microbiota would, in our view, fall somewhat outside the scope of the current study. We have no direct research question or mechanistic model in mind (e.g., none of the migrant populations were war-refugees), and we felt such analyses would remain explorative at this stage.

Can you further define living in the same urban area? How large is that urban area?

The urban area is the municipality of Amsterdam. The city proper is a moderate sized (219.49 km²) with approximately 900,000 inhabitants, and is the capitol of the Netherlands.

In response we have adapted (*in italic*) the introduction on page 6 text as follows:

The HELIUS (Healthy Life in an Urban Setting) study is a multi-ethnic cohort study among citizens of Amsterdam, The Netherlands^{19,20}. *“The city proper is a moderately sized area (219.49 km²) with approximately 900,000 inhabitants, and is the national capitol.”*

You also need to present more background on your prior work with differences in depression

among these groups to understand your hypothesis better. You note that depression is heterogenous but then do not really look at it beyond the PHQ-9.

We thank the reviewer for this comment and the opportunity to clarify. The statement that depression is an heterogenous construct was not intended to forecast a research question, but aimed to provide a further explanation for why previously observed associations tend to be modest in size and vary between study populations. This argument lead us to conclude that “adequately powered studies” are needed (on page 5). Notwithstanding, depression subcategories exist and these may differentially impact biology, which poses a potential limitation to the interpretation of our findings and a relevant objective for future studies. We have now added this point to the discussion on page 15:

“Depression is a heterogenous construct whereby subtypes have been identified by symptom profile. This aspect warrants further research in light of evidence that these subtypes may exert differential biological effects (Milaneschi, 2016; Penninx, 2016).”

In the Intro you bring up that the literature has sparse adjustments for factors such as lifestyle. you have attempted to include factors such as exercise (although how accurate is minutes per week), smoking, alcohol use but does not really account for diet (beyond metabolic factors such as HbA1C.

The author is absolutely correct to suggest that the number of factors one might take into account could be vastly expanded, e.g., to include diet parameters, additional risk factors of depression, environmental or lifestyle factors tied to particular ethnic groups, etc. There are conceptual and methodological reasons, in our view, that favor to limit such expansion of the covariate set in the absence of a sound mechanistic view on how such covariates operate. We therefore focused on an a priori selection based on the extant epidemiological literature. And even within this more conservative approach we might be at risk of over-correcting; this is something we also addressed as a caveat in the discussion (on page 15), i.e., when mentioning that adjusted results might be potentially misleading when a covariate is on the causal pathway.

To illustrate this concern with a hypothetical example: if, for instance, we would find that the statistical association between depression and microbiome weakens after adjustment for a particular dietary factor, e.g., sugar consumption, then such finding would afford multiple interpretations: For example, depression-related sugar intake may have depressogenic effects via effects on the microbiota (e.g., by inducing particular metabolites), or the depression-associated microbiota affects sugar craving/intake (whereby not the microbiota but sugar is the ‘real’ depressogenic factor), or depression-associated sugar consumption affects microbiota composition (whereby the latter may or may not have a mechanistic impact), or by another pathway, or a combination of those pathways. We feared that selecting covariates on the basis of plausible assumptions, but without a mechanistic underpinning, may ultimately raise more questions than give answers. Additionally it increases the risk of spurious/accidental findings. A more pragmatic reason is that we only have full dietary analyses for a part of our sample. We therefore did not opt for further exploratory analyses in the current paper.

A small thing, but be consistent throughout with things like PPI versus proton pump inhibitor--PPI used before proton pump inhibitor spelled out.

Thanks for noting: this has been corrected.

For methods, the first twenty principal coordinates were used--often this many are not used. What was the rationale?

The reviewer is correct and six coordinates were retained in the final model. These coordinates were selected (from among the first 20) for their ability to significantly predict depressive symptom levels, using forward selection, yielding a sparse and nonredundant parameter set. The text in the results has now been adapted to further clarify that.

Results, page 11: *“Forward selection of the first 20 coordinates yielded 6 coordinates that compiled information predictive of depressive symptom scores, and these coordinates were used in subsequent regression analyses.”*

My understanding is that Greengenes has not been updated in multiple years and this is a limitation of it.

We agree with the reviewer, and Greengenes is less preferred in our view also. However, it is still frequently used in the relevant literature, and dominates the older literature. Hence, to facilitate comparison of our findings with the extant body of literature, we opted to report the results using both Greengenes and SILVA.

The discussion provides an example whereby using both data bases allowed us to identify, and explain, findings that might otherwise have added confusion to the literature, (page 15, first paragraph). *“For example, the sharply heterogeneous pattern of associations with depression among 10 ASVs that Greengenes maps onto the single genus *Oscillospira* disappeared when utilizing the Silva database, which assigned these 10 ASVs to 8 different genera.”*

Your conclusion is not clear. You state the associations are largely invariant but that the microbiota may partly account for depression disparities.

Thanks for this critical point: the reviewer seems to suggest (and we apologize if we misinterpreted) that the observation that associations are largely invariant across ethnic groups might be inconsistent with the conclusion that the microbiota could partly account for depression disparities? This assumption would not be correct in our view. We may illustrate this with hypothetical example whereby we replace ‘microbiota’ with ‘health’’: If a more-or-less universal association between health and mental wellbeing is observed (i.e., the size of this association is comparable across ethnicities) than that would not contradict that the ethnic disparities in health may account for disparities in mental well-being. In fact, it likely is the opposite: if the association between health and wellbeing differs between ethnicities, then that would weaken (not strengthen) the probability that health disparities predict well-being between ethnic groups.

Neuroticism should have been brought up sooner as you say alpha-diversity no longer differed when also including neuroticism. What is the take-home message? What are next steps?

In response we have adapted the introduction to better clarify the reasons for including neuroticism in auxiliary analyses, on page 5.

“For most individuals depression is transient with a median duration of three to six months. Auxiliary analyses took pre-existing markers of depression risk into account, as these may provide a window on the temporal specificity of associations between the microbiota and current symptom levels; these included prior depressive episodes, parental history of depression, and the personality trait neuroticism (a generic risk marker for psychopathology).”

Based on the observation that the association with alpha-diversity disappeared after adjusting for neuroticism, we proposed in the discussion that the microbiota-depression association may involve both generic and depression-specific mechanisms (discussion, bottom of page 13)

REVIEWERS' COMMENTS

Reviewer #1 (Remarks to the Author):

The authors have satisfactorily addressed all my points, and I have no further comments.